# Anti-Human PD-L1 Nanobody for Immuno-PET Imaging: Validation of a Conjugation Strategy for Clinical Translation

**DOI:** 10.3390/biom10101388

**Published:** 2020-09-29

**Authors:** Jessica Bridoux, Katrijn Broos, Quentin Lecocq, Pieterjan Debie, Charlotte Martin, Steven Ballet, Geert Raes, Sara Neyt, Christian Vanhove, Karine Breckpot, Nick Devoogdt, Vicky Caveliers, Marleen Keyaerts, Catarina Xavier

**Affiliations:** 1Medical Imaging Department (MIMA), In Vivo Cellular and Molecular Imaging Laboratory (ICMI), Vrije Universiteit Brussel, Laarbeeklaan 103, Building K, 1090 Brussels, Belgium; pieterjan.debie@vub.be (P.D.); nick.devoogdt@vub.be (N.D.); vicky.caveliers@vub.be (V.C.); marleen.keyaerts@vub.be (M.K.); Catarina.xavier@vub.be (C.X.); 2Department of Biomedical Sciences, Laboratory for Molecular and Cellular Therapy (LCMT), Vrije Universiteit Brussel, Laarbeeklaan 103, Building D, 1090 Brussels, Belgium; Katrijn.Broos@vub.be (K.B.); Quentin.Lecocq@vub.be (Q.L.); karine.breckpot@vub.be (K.B.); 3Research Group of Organic Chemistry (ORGC), Vrije Universiteit Brussel, Pleinlaan 2, 1050 Brussels, Belgium; charlotte.martin@vub.be (C.M.); Steven.Ballet@vub.be (S.B.); 4Sciences and Bioengineering Sciences, Cellular and Molecular Immunology laboratory (CMIM), Vrije Universiteit Brussel, Pleinlaan 2, Building F, 1050 Brussels, Belgium; geert.raes@vub.be; 5Myeloid Cell Immunology Laboratory (MCI), VIB Inflammation Research Center, Technologiepark-Zwijnaarde 71, 9052 Ghent, Belgium; 6MOLECUBES NV, Ottergemsesteenweg Zuid 325, 9000 Ghent, Belgium; sara.neyt@molecubes.com; 7IBiTech-MEDISIP, Ghent University Hospital Site, Block B, Corneel Heymanslaan 10, 9000 Ghent, Belgium; Christian.Vanhove@UGent.be; 8Nuclear Medicine Department, UZ Brussel, Laarbeeklaan 101, 1090 Brussels, Belgium

**Keywords:** Nanobody, PD-L1, site-specific, PET, gallium-68, Sortase A, cancer

## Abstract

Immune checkpoints, such as programmed death-ligand 1 (PD-L1), limit T-cell function and tumor cells use this ligand to escape the anti-tumor immune response. Treatments with monoclonal antibodies blocking these checkpoints have shown long-lasting responses, but only in a subset of patients. This study aims to develop a Nanobody (Nb)-based probe in order to assess human PD-L1 (hPD-L1) expression using positron emission tomography imaging, and to compare the influence of two different radiolabeling strategies, since the Nb has a lysine in its complementarity determining region (CDR), which may impact its affinity upon functionalization. The Nb has been conjugated with the NOTA chelator site-specifically via the Sortase-A enzyme or randomly on its lysines. [^68^Ga]Ga-NOTA-(hPD-L1) Nbs were obtained in >95% radiochemical purity. In vivo tumor targeting studies at 1 h 20 post-injection revealed specific tumor uptake of 1.89 ± 0.40%IA/g for the site-specific conjugate, 1.77 ± 0.29%IA/g for the random conjugate, no nonspecific organ targeting, and excretion via the kidneys and bladder. Both strategies allowed for easily obtaining ^68^Ga-labeled hPD-L1 Nbs in high yields. The two conjugates were stable and showed excellent in vivo targeting. Moreover, we proved that the random lysine-conjugation is a valid strategy for clinical translation of the hPD-L1 Nb, despite the lysine present in the CDR.

## 1. Introduction

Immune checkpoints, which mitigate immune activities under physiological conditions, can be hijacked by cancer cells, thereby overcoming the patient’s anti-tumor immune response. Amongst the immune checkpoints, programmed death-ligand 1 (PD-L1) and its receptor programmed death-1 (PD-1), have drawn the most attention during the last 10 years and monoclonal antibody (mAb)-based immunotherapies antagonizing PD-1/PD-L1 have revolutionized the cancer treatment paradigm [1,2]. Recent clinical trials showed long-lasting responses and provided evidence for their importance in cancer therapy [3]. However, only a subset of patients benefits from existing treatments, typically 20% [4]. This low response rate can be explained, at least partially, by the lack of a precise technique to select patients with PD-L1 expression [5,6]. Immunohistochemistry (IHC) is currently employed in order to assay PD-L1 expression by staining a biopsy sample, which is not representative for the heterogeneous expression within the primary tumor lesion, nor within the metastases [7]. In addition, PD-L1 expression within the tumor microenvironment is a dynamic process that can be influenced by routine cancer treatments, such as radiotherapy [8]. Moreover, PD-L1 expression is not limited to tumor cells, as immune cells within the tumor can also express PD-L1 and contribute to immune escape [9]. To date, several antibodies are available for IHC with variable ability to stain PD-L1 on tumor cells and immune cells, implying that the pathologist’s interpretation during analysis is important to categorize samples as positive or not [10].

Pre-clinical and clinical studies with radiolabeled mAbs showed that the assessment of the PD-L1 status by positron emission tomography (PET) imaging is a promising strategy for patient stratification [11,12,13]. However, mAbs have very slow clearance kinetics from the blood through the hepatobiliary system and, therefore, imaging is performed several days post-injection (p.i.), in order to achieve a sufficiently low background signal to visualize the molecular target.

Nanobodies (Nbs), also called single domain antibody fragments, can overcome limitations linked to the use of mAbs for PET imaging. Their high affinity and specificity, as well as their small size (~15 kDa), make Nbs ideal probes for molecular imaging. Their fast blood clearance allows for imaging as early as 1 h p.i. and radiolabeling with short-lived radioisotopes, lowering the radiation burden for the patients [14]. We previously selected a lead high affinity PD-L1-specific Nb that allowed non-invasive SPECT/CT imaging of PD-L1 expression in murine tumor models with varying PD-L1 expression. [15] The ^99m^Tc-labeled Nb revealed high signal-to-noise ratios, strong ability to specifically detect PD-L1 in melanoma and breast tumors, and surprisingly low kidney retention, unlike many other radiolabeled Nbs.

For further clinical translation, we aimed to radiolabel and validate the Nb with a PET radioisotope, such as Gallium-68 (^68^Ga) (radioactive half-life 67.7 min.), which is commonly used in clinical settings. To allow for chelation of ^68^Ga, Nbs are typically conjugated with the bifunctional chelating agent 2-*S*-(4-isothiocyanatobenzyl)-1,4,7-triazacyclononane-1,4,7-triacetic acid (*p*-NCS-Bn-NOTA) on the primary amino groups of the lysines of the Nb’s structure [16]. Although this strategy is straightforward and already applied to functionalize Nbs undergoing clinical trials [16,17], it can present a problem when there are lysines in one of the three complementarity determining regions (CDRs) of the Nb sequence, as is the case for the PD-L1-specific lead compound [18], as these are hypervariable regions that are likely involved in antigen-binding.

In the current study, we aimed to determine whether the lysine in the CDR could hamper binding capacity of the Nb, which is critical information before selecting a bioconjugation technique suitable for clinical translation. To this end, we compared the impact on the Nb of two bioconjugation strategies: a classical approach using conjugation on lysines (further referred to as random approach) versus a site-specific approach while using the Sortase-A-mediated transpeptidation [19,20].

## 2. Materials and Methods

### 2.1. Reagents

All of the reagents and solvents were purchased from Sigma–Aldrich (Overijse, Belgium) or VWR (Oud-Heverlee, Belgium). Buffers used for coupling reactions and for radiolabeling were prepared with metal free water (Honeywell, Fluka, Brussels, Belgium) and purified from metal contamination using Chelex 100 resin. *p*-SCN-Bn-NOTA was purchased from Macrocyclics (Plano, USA). ^68^Ga was obtained from a ^68^Ge/^68^Ga generator (Galli Eo™, IRE ELiT, Fleurus, Belgium) eluted with 0.1 N HCl. [^67^Ga]Ga-citrate solution was purchased from Mallinckrodt (Amsterdam, The Netherlands).

### 2.2. Chromatographic Analysis

Size-exclusion chromatography (SEC) columns were purchased from GE Healthcare (Diegem, Belgium). SEC purification of the site-specifically functionalized Nb was performed on a Superdex 75 Increase 10/300 GL column using 0.1 M NH_4_OAc Ph 7, at a flow rate of 0.8 mL/min. SEC purification of randomly functionalized Nb was performed on a Superdex Peptide 10/300 GL column while using 0.1 M NH_4_OAc pH 7, at a flow rate of 0.5 mL/min. For quality control (QC), radiochemical purity (RCP) was assayed with binderless glass microfiber paper that was impregnated with silica gel (instant thin layer chromatography, iTLC-SG) (Agilent Technologies, Diegem, Belgium) using 0.1 M sodium citrate buffer pH 4.5–5 as eluent. RCP was also assayed by SEC on a Superdex Peptide 3.2/300 GL using a 2× Phosphate-buffered saline (2× PBS: 5.36 mM KCl, 273.8 mM NaCl, 2.94 mM KH_2_PO_4_, 16.2 mM Na_2_HPO_4_) at a flow rate of 0.150 mL/min. Serum and urine samples were analyzed on a Superdex 5/150 GL using 2× PBS at a flow rate of 0.3 mL/min.

### 2.3. Production and Purification of the hPD-L1 Nb and Sortase-A Enzyme

The lead hPD-L1 Nb K2 with a his_6_-tag and/or a sortag (LPETG) and the Sortase-A enzyme were produced as described before [15,19,20].

### 2.4. Synthesis of GGGYK-NHCS-Bn-NOTA

The peptide (Appendix A) was synthesized by manual solid phase peptide synthesis (SPPS), as described in the Appendix A.

### 2.5. Site-Specific Nb Functionalization

These procedures are essentially described elsewhere [20]. To the (hPD-L1)-sortag-his_6_-tag Nb (1 eq., 50 μM in final volume) in Tris-buffered Saline (TBS) pH 7, was added GGGYK-NHCS-Bn-NOTA (20 eq., 1 mM in final volume) and his_6_-tagged Sortase-A enzyme (2 eq., 100 μM in final volume). 150 µL (10% of final reaction volume) of 10× Sortase buffer (500 mM Tris-HCl, 150 mM NaCl, 100 mM CaCl_2_, pH 7.7) was added and the reaction volume was topped up to 1.5 mL with metal free water. The reaction mixture (RM) was incubated 16 h at 37 °C. The his_6_-tagged compounds (unreacted Nb, Sortase-A enzyme, cleaved *C*-terminus of the Nb) were removed by immobilized metal affinity chromatography (IMAC) by adding Ni-NTA resin (500 µL in TBS, Thermo Fisher Scientific, Belgium) and then moderately shaken for 2 h at room temperature (RT). After filtration, the flow-through was incubated with excess EDTA (50 mM final concentration) for 1 h at RT. RM was concentrated on a vivaspin 2 (MWCO 5KDa, Sartorius, Belgium) and then purified by SEC.

### 2.6. Random Conjugation of the NOTA Chelator to the Lysines of the Nb

Conjugation was performed, as described elsewhere [16]. Briefly, the his_6_-tag hPD-L1 Nb in 0.05 M sodium carbonate buffer (1.2 mg/mL, 2 mL) was incubated for 2.5 h at RT with *p*-NCS-Bn-NOTA (20 eq.) at pH 8.5–8.7. RM was neutralized by the addition of 1 N HCl, concentrated, and purified by SEC.

### 2.7. Quality Controls of NOTA-Nbs

The purity of the functionalized hPD-L1 Nbs was assessed by SEC, SDS-PAGE, and Western Blot (WB). The equilibrium dissociation constant (K_D_) of the unmodified and NOTA-functionalized Nbs was measured by surface plasmon resonance SPR. The procedures are described in the Appendix A.

### 2.8. Nanobody Radiolabeling

The site-specifically or randomly conjugated NOTA-(hPD-L1) Nb (7.0–9.0 nmol) in 1 mL of 1 M NaOAc buffer pH 5 was incubated 10 min. at RT with 1 mL of ^68^Ga eluate (340-950 MBq). The radiolabeled-Nb solution was purified by SEC on a PD-10 column (GE Healthcare, Belgium) pre-equilibrated with freshly prepared 0.9% NaCl containing 5 mg/mL vitamin C pH 5.8–6.1 (injection buffer). The final solution was filtered through a 0.22 µm filter (Millipore, Belgium). RCP was determined before and after purification by radio-iTLC ([^68^Ga]Ga-NOTA-Nb Rf = 0, [^68^Ga]Ga-citrate Rf = 1).

For studies that require later time point analysis, labeling with Gallium-67 (^67^Ga, t_1/2_ = 78.3 h) was performed. The NOTA-(hPD-L1) Nb solution (7.0 nmol) was brought to pH 5 with 5 M NH_4_OAc pH 5–5.2 and incubated for 10 min. at RT with 200 μL (100–400 MBq) of [^67^Ga]GaCl_3_, obtained from [^67^Ga]Ga-citrate, as previously described [21]. The radiolabeled Nb was purified by SEC NAP-5 column (GE Healthcare, Belgium), eluted with 1 mL of injection buffer, and filtered. RCP was determined by radio-iTLC. 

Decay-corrected radiochemical yield (DC-RCY) was calculated for the time point after SEC purification.

### 2.9. Stability Studies

The stability of the ^68^Ga or ^67^Ga-labeled Nbs (15–50 MBq, after filtration) was tested over 4 h at RT, in human serum (HS) at 37 °C, and in the presence of a 1000-fold excess of competitor (DTPA chelator, which is able to chelate ^68^Ga at RT) to assay transchelation that could occur with transferrin in vivo before going to mouse models. At different time points, the aliquots were analyzed by radio-iTLC and radio-SEC.

### 2.10. Animal Models and Cell Lines 

Dr. S.L. Topalian (National Cancer Institute, USA) provided HLA-A*0201^+^ 624-MEL cells. The 624-MEL cells were stably transduced to express hPD-L1 and they have been characterized, as previously described [15]. The 624-MEL cells were cultured in RPMI1640 medium supplemented with 10% Fetal clone I serum (Thermo Scientific, Belgium), 2 mM L-Glutamine, 100 U/mL penicillin, 100 µg/mL streptomycin, 1 mM sodium pyruvate, and nonessential amino acids. Female, five to six weeks old C57BL/6 mice (for biodistribution, biological half-life in blood and stability studies), and athymic nude Crl:NU(NCr)-Foxn1nu mice (for tumor targeting) were purchased from Charles River. All of the experiments were performed in accordance with the European guidelines for animal experimentation under the license LA1230272. Experiments were approved by the Ethical Committee for the use of laboratory animals of the Vrije Universiteit Brussel (17-272-6 and 20-272-4). Intravenous injections were performed in the tail vein. The animals were anesthetized with 2.5% isoflurane in oxygen (Abbott) for injections, samplings, imaging, and euthanasia.

### 2.11. Cell Binding Study

The radiolabeled Nb’s specificity was tested on transduced hPD-L1 positive (hPD-L1^POS^) 624-MEL cells [15]. Excesses of unmodified, unlabeled Nb, or untransduced hPD-L1 negative (hPD-L1^NEG^) 624-MEL cells were used as control conditions. The procedures are detailed in the Appendix A.

### 2.12. Affinity Measure (K_D_) by Cell Saturation Assay

The affinity of the radiolabeled Nb was tested on hPD-L1^POS^ 624-MEL cells. 5x10^4^ cells in 1 mL of medium per well were allowed to attach in a 24 well plate at 37 °C two days prior to the experiment. The plate was cooled to 4 °C 1 h prior to the experiment. The supernatant was removed and the cells were incubated for 1 h at 4 °C with 500 μL of a ^68^Ga-labeled Nb solution at different concentrations (300 nM, 100 nM, 33.3 nM, 11.1 nM, 3.7 nM, 1.2 nM, 0.4 nM, and 0.1 nM) in unsupplemented medium (N = 3 wells per condition). The wells were processed the same way as the cell binding study. To correct for nonspecific binding, the same procedure was simultaneously applied to a second plate containing 100-molar excess of unlabeled competitor in each well. The K_D_ was calculated while using a “One site—total and nonspecific binding” analysis in Prism software.

### 2.13. Biological Half-Life in Blood of the ^68^Ga-Labeled Nbs

C57BL/6 mice were injected intravenously (N = 6 per group) with 6.6 ± 1.2 MBq of randomly or site-specifically ^67^Ga-labeled Nb (10 μg NOTA-Nb) and blood samples from different time points (5, 10, 20, 30, 40, 60, 120, and 180 min.) were counted against a standard of known activity using a γ-counter. Blood sample volume was calculated and activity in blood was expressed as a percentage of injected activity per total blood volume (%IA/TBV). The biological half-life was calculated using a one phase decay model in the Prism software. The experiment was repeated with 28.6 ± 2.1 MBq of ^68^Ga-labeled Nbs (10 μg NOTA-Nb).

### 2.14. In Vivo Stability Studies

C57BL/6 female mice (N = 8 per group) were intravenously injected with 9.4 ± 1.2 MBq of randomly or site-specifically ^67^Ga-labeled Nb (10 μg NOTA-Nb). At each time points (5, 15, 45, and 120 min.), two animals were euthanized to collect blood and urine samples to determine the percentage of intact ^67^Ga-labeled Nb in the samples. Samples were diluted with 0.1 M sodium citrate buffer pH 4.5–5 containing 0.1% Tween 80, filtered with 0.22 μm filter and analyzed by radio-SEC. The experiment was repeated with ^68^Ga-labeled Nbs with the same experimental settings (10 μg NOTA-Nb, 30.5 ± 7.1 MBq).

### 2.15. Biodistribution and Comparative Tumor Targeting Studies

Athymic nude mice (seven weeks old, 42 animals) were subcutaneously injected in the right leg with 5x10^6^ hPD-L1^POS^ 624-MEL cells. Tumor volume was measured twice weekly with an electronic caliper and calculated using the following formula: (length × width^2^)/2. The animals were randomized (N = 21/group, two groups). After three to six weeks post inoculation, the tumor volume reached 99 ± 54 mm^3^ for the first group and 87 ± 44 mm^3^ (NS) for the second group, injected respectively with site-specifically ^68^Ga-labeled Nb (4.5 µg NOTA-Nb; 20.5 ± 2.1 MBq, 64.8 GBq/μmol) and randomly ^68^Ga-labeled Nb (6.0 μg NOTA-Nb; 19.4 ± 2.1 MBq, 45.8 GBq/μmol). Injected activities and apparent molar specific activities are reported for the time of injection.

As a control group, athymic nude mice (six weeks old, N = 6/group) were injected with 4.2 × 10^6^ hPD-L1^NEG^ 624-MEL cells, allowing for reaching 303 ± 303 mm^3^ in three weeks.

Biodistribution was evaluated at 1 h 20 p.i. After euthanasia, main organs and tissues were isolated, weighed, and counted against a standard of known activity using a γ-counter. The amount of radioactivity in organs and tissues was expressed as percentage of injected activity per gram (%IA/g), corrected for decay. A single cell suspension from the tumors was prepared and flow cytometry analysis was performed in order to characterize hPD-L1 expression (procedure in the Appendix A).

### 2.16. PET/CT Iimaging and Analysis

hPD-L1^POS^ 624-MEL xenografted athymic nude mice (nine weeks old), with a tumor size of (615 ± 502) mm^3^ (N = 2) were injected with site-specifically ^68^Ga-labeled Nb (11 μg NOTA-Nb, 17.8 ± 2.2 MBq, 23.0 GBq/µmol). The acquisition was performed with a β-CUBE PET/CT system (MOLECUBES, Ghent, Belgium) 1 h 20 min. p.i. Total PET/CT scanning time was 17 min. The PET images were acquired over 15 min. and reconstructed into a matrix of 193 × 192 × 384 voxels with 400 μm voxel size. The CT images were iteratively reconstructed using the ISRA reconstruction algorithms into 200 μm voxels (matrix 200 × 200 × 393). The animals injected with randomly ^68^Ga-labeled Nb or animals bearing negative tumors were not scanned for logistical reasons.

### 2.17. Statistical Analyses

The calculation of the amount of necessary animals for the comparative tumor targeting study between randomly and site-specifically labeled Nbs was performed using a Wilcoxon–Mann–Whitney test (two groups) analysis in G POWER (considering that 1% difference uptake would be a relevant difference between the two groups for this model, with 95% confidence, with a pooled standard deviation (stdev) of 1.09087 based on preliminary targeting studies. 

The results are expressed as mean ± stdev. A non-parametric Mann–Whitney U test was carried out to compare the data sets. Sample sizes and number of repetitions of experiments are indicated in the figure legends or in the materials and methods section. The number of asterisks in the figures indicates the statistical significance, as follows: * *p* < 0.05; ** *p* < 0.01; *** *p* < 0.001; Non-significant (NS).

## 3. Results 

### 3.1. Nanobody Functionalization and Affinity Analysis

To allow for PET-imaging with a lead Nb targeting human PD-L1, both a random and site-specific coupling with NOTA chelator was performed. The site-specific approach not only has the advantage of producing a homogenous end product [19,20], but is also important for this particular Nb, as the Nb contains besides the *N*-terminal amine and three lysines in the framework region also a lysine in one of the CDR regions, which could result in decreased binding affinity upon random conjugation with amine-targeted chemistry. The site-specific coupling was performed by an enzymatic reaction, which involves an overnight incubation of the Sortase-A enzyme with a hPD-L1 Nb containing the Sortase-A recognition site LPETG (sortag) at the *C*-terminus and a substrate. The substrate consists of a NOTA-chelator that is covalently linked with a triglycine sequence, acting as a nucleophile attacking the Nb-Sortase-A intermediate. The site-specifically modified Nb was obtained in 56 ± 4% yield (N = 3). After purification, a purity of >95% was measured with SEC (Appendix A), SDS-PAGE, and western blot (Appendix A). The random strategy, involving thiourea formation with *p*-SCN-Bn-NOTA on solvent-exposed lysines, yielded 52 ± 3% (N = 3) of functionalized hPD-L1 Nb, with >93% purity, as measured on SEC (Appendix A). The site-specific conjugation resulted in a homogenous product of Nb coupled to one NOTA, while the random conjugation reaction resulted in a mixture mainly containing Nbs with zero or one NOTA chelators, as determined by ESI-Q-ToF-MS analysis (Appendix A).

The affinity kinetics with conjugated and unconjugated Nbs were measured by SPR on immobilized hPD-L1 recombinant protein; Both the site-specifically and randomly modified NOTA-Nbs, exhibited a K_D_ in the same range as the unconjugated Nb precursors (Table 1), which suggested that there is no impact on the functionality of the hPD-L1 Nb after NOTA-conjugation.

### 3.2. Radiolabeling and In Vitro Stability Studies 

Site-specifically and randomly labeled [^68^Ga]Ga-NOTA-(hPD-L1) Nbs were obtained with both a RCP >95%, and DC-RCY of 82 ± 2% (N = 5) and 75 ± 1% (N = 4), respectively, in a procedure time of 30 min. The respective apparent molar specific activities, after PD10 purification, were 66.8 ± 1.4 GBq/μmol (N = 5) and 51.7 ± 3.5 GBq/μmol (N = 4) for site-specifically and randomly labeled Nbs, respectively. 

Besides labeling with ^68^Ga (t_1/2_ = 67.7 min.), we also performed labeling with ^67^Ga (t_1/2_ = 78.3 h) to allow prolonged testing. Site-specifically and randomly labeled [^67^Ga]Ga-NOTA-(hPD-L1) Nbs were obtained with a RCP > 99% and DC-RCY were 87 ± 5% (N = 3) and 86 ± 10% (N = 3), respectively, for a procedure time of 30 min.

In vitro stabilities of the radiolabeled probes are summarized in Table 2. In injection buffer at RT, the RCP of the site-specifically ^68^Ga-labeled Nb remained > 99% after 4 h. After 4 h at 40 °C in injection buffer, 6% of a radiolysis side product was observed, showing an influence of temperature on the radiolysis process.

In the presence of 1000-molar fold excess DTPA, no transchelation was observed and the RCP of the site-specifically labeled [^68^Ga]Ga-NOTA-(hPD-L1) Nb remained > 98% after 4 h of incubation.

In human serum at 37 °C, both ^68^Ga-labeled Nbs were stable over 1 h (RCP > 95%); however, some degradation occurred at later time points, resulting in 80% of Stable 6^8^Ga-labeled Nbs after 4 h of incubation (for the site-specific conjugate). In the same conditions, the site-specifically ^67^Ga-labeled Nb was stable over 4 h (RCP > 95%). In both cases, control sample diluted in PBS at 37 °C (data not shown) showed similar profiles as in human serum, showing that the degradation is rather related to temperature than to the serum condition, as already observed in other studies [22]. 

### 3.3. In Vitro Binding Capacity on Cells 

In order to assess the specificity of the site-specifically and randomly coupled Nbs to bind to hPD-L1 expressed on cells, they were labeled with ^68^Ga and added to either hPD-L1^POS^ or hPD-L1^NEG^ 624-MEL cells. After incubation, the unbound fraction was washed away and the cell-associated activity was measured. The % of cell-associated activity of both radiolabeled Nbs showed specific binding on hPD-L1^POS^ cells, which was confirmed by absence of cell-associated activity in control conditions (hPD-L1^NEG^ cells and excess of unlabeled Nb, Figure 1). For both site-specifically and randomly labeled Nbs, a significantly higher amount of bound activity on hPD-L1^POS^ cells was measured than on hPD-L1^NEG^ cells (site-specifically-labeled; 13.5 ± 4.9% vs. 0.4 ± 0.1%, respectively; *p* < 0.0001; randomly-labeled; 3.0 ± 1.4% vs. 0.3 ± 0.1%, respectively, *p* < 0.0007). This assay confirms the specificity of the hPD-L1 Nbs for their target, but does not allow assessing affinity.

### 3.4. Affinity Assay (K_D_) by Cell Saturation

The K_D_ calculated from SPR was similar for both randomly and site-specifically functionalized Nbs (non-radiolabeled compounds), as shown above. The K_D_ was calculated from a cell saturation assay using the ^68^Ga-labeled probes to investigate the affinity of the radiolabeled Nbs (Figure 2). When considering the potential error margins on this experiment (on the number of cells, on the probe dilutions), the two values are considered in the same range (0.8 nM for the randomly labeled Nb, 1.2 nM for the site-specifically labeled Nb) and are in the usual range for high affinity Nbs [23].

### 3.5. Biological Half-Life in Blood and In Vivo Stability Studies 

The biological half-lives in blood were 13.8 ± 2.0 min. and 12.2 ± 2.0 min. (NS) for the site-specifically and randomly radiolabeled Nb, respectively (Figure 3). Both Nbs exhibit a typical Nb clearance profile with a fast initial clearance phase and a slower second clearance phase.

Up to 15 min., both the radiolabeled Nbs remained intact in plasma (>99% of activity was intact Nb). At later time points, activity in plasma was too low to allow for analysis, even with ^67^Ga-labeled Nbs. In urine, analyses up to 120 min. revealed >90% of intact excreted site-specifically radiolabeled Nb compared to only 70% for the randomly radiolabeled Nb (Appendix A).

### 3.6. Biodistribution, In Vivo Tumor Targeting and PET/CT Imaging

Appendix A summarizes the biodistribution in C57BL/6 mice of site-specifically and randomly labeled [^68^Ga]Ga-NOTA-(hPD-L1) Nbs. For both probes, ex vivo analysis 1 h 20 p.i. showed very low uptake in all organs, except in the kidneys due to renal excretion. Notably, retention in the kidneys of the site-specifically labeled hPD-L1 Nb was 10.1 ± 2.4%IA/g, which is, to our knowledge, the lowest ever reported for a radiolabeled Nanobody at an early time-point.

Biodistribution and tumor targeting in athymic nude mice bearing hPD-L1^POS^ cells, or hPD-L1^NEG^ cells as a control (preliminary studies, Figure 4a, data in Appendix A) showed specific accumulation in the hPD-L1^POS^ tumor; about six times higher (*p* < 0.0001) than in the hPD-L1^NEG^ tumors for the site-specifically radiolabeled Nb, and about five times higher (*p* < 0.0001) for the randomly radiolabeled Nb. Both probes showed high uptake variation in the hPD-L1^POS^ tumors, which did not allow to conclude on a potential difference in affinity in vivo between the two radiolabeled Nbs. Using these preliminary data and taking into account variations in tumor size, a larger sample of animals (N = 21/group) was calculated to be necessary in order to make a statistically reliable conclusion. For both probes, the accumulation of radioactive signal in the hPD-L1^POS^ tumors was measured: 1.89 ± 0.40%IA/g for the site-specifically radiolabeled Nb and 1.77 ± 0.29%IA/g for the randomly radiolabeled Nb (P = 0.263, NS), as depicted in Figure 4b.

The tumor-to-blood (T/B) ratio and tumor-to-muscle (T/M) ratio were calculated for both probes. T/B ratio for the site-specific Nb was 6.3 ± 3.0 (N = 21) and 5.4 ± 1.5 (N = 21) for the random Nb (NS). The T/M ratio for the site-specific Nb was 34.5 ± 13.2 (N = 21) vs. 28.0 ± 10.6 (N = 21) for the random Nb (NS).

Ex vivo flow cytometry characterization of the hPD-L1^POS^ tumor cells confirmed the significantly higher (*p* < 0.0001) % of cells expressing hPD-L1 than the hPD-L1^NEG^ tumor cells, as reported in Appendix A. 

The PET/CT image of the animal bearing a hPD-L1^POS^ tumor and injected with site-specifically [^68^Ga]Ga-NOTA-(hPD-L1) Nb confirms the low background at 1 h 20 p.i. and high uptake in the tumor lesion (Figure 4c).

## 4. Discussion

In the past years, antibody-based treatments blocking the interactions between PD-1 and its ligand PD-L1 have been developed to restore the patient’s anti-cancer immune activity, however providing clinical benefits in only a fraction of all patients. Many novel PET imaging agents are being developed in order to improve upfront patient selection as well as follow up changes in PD-L1 expression during treatments, based on full antibodies or smaller targeting moieties, such as peptides [24], affibodies [25], adnectins [26], and Nanobodies [27,28]. Here, we present our results on a Nb targeting human PD-L1 that was previously selected based on its theranostic capacity [15], and that we now prepare for patient use by the development of ^68^Ga labeling strategies suitable for clinical applications.

The hPD-L1 Nb was functionalized with NOTA in a site-specific manner using an enzymatic coupling reaction at the *C*-terminal end of the Nb and in a random manner on exposed lysine residues throughout the Nb protein. The Nb protein contains in total four lysines, including a lysine residue located in one of its binding region. It is unknown which lysine or how many lysines will be coupled to NOTA after a random coupling, and thereby this strategy could affect the Nb’s binding potential. Both strategies resulted in good recovery yields, high radiochemical yields and excellent radiochemical purity. However, the site-specific strategy allowed ^68^Ga-labeling with less Nb, resulting in an injectable product with higher apparent molar specific activity than for the random strategy.

Both radiolabeled Nbs were stable in vitro in injection buffer over 4 h. In human serum at 37 °C, both ^68^Ga-labeled Nbs were stable over 1 h (>95% RCP). A smaller fragment could be observed at later time points (4 h) and it is most likely due to radiolysis, which is a result of combined positron emission and high temperatures [22]. As the biological half-life of the compounds is very short and since the product is diluted in the blood, radiolysis effects will be lower, and these effects will not be clinically relevant. 

The stability of both probes was also evaluated in vivo up to 2 h post injection. The radioactive metabolites, as observed in urine, were relatively higher for the random than for the site-specific compound, but no signs of recirculation in the blood could be detected, which makes those metabolites not relevant for imaging purposes.

In vitro and in vivo, the site-specific strategy could theoretically provide better targeting properties, since the attached chelator does not interfere with the CDRs interacting with the antigen. Experimental data however demonstrated that tumor uptake of the randomly radiolabeled Nb is as good as the site-specifically radiolabeled analogue.

When considering that the lead compound hPD-L1 Nb was found to be not cross-reactive to the murine PD-L1 [15], no specific uptake in other organs or tissues (besides the tumor) in mice was expected. In patients, the biodistribution of the hPD-L1 Nb in healthy tissues will have to be assessed. Uptake can mainly be expected in the spleen as for the ^18^F-labeled Adnectin [12] and brown fat as reported for the murine PD-L1 binder in wild mice [29]. Kidney and bladder retention are due to the excretion route of Nbs. Kidney retention of the randomly labeled Nb was higher than for the site-specifically radiolabeled Nb, which can be attributed to the presence of a his_6_-tag at the *C*-terminus of the randomly radiolabeled Nb, affecting the *C*-terminal charge as already observed [30,31], while the his_6_-tag is removed by the Sortase-A enzyme during the site-specific coupling [32]. Nevertheless, kidney uptake of the site-specifically radiolabeled hPD-L1 Nb is the lowest ever reported for a Nb. The rationale for this effect is still unknown and could be further investigated in the future.

Preliminary studies involving six animals per group bearing hPD-L1^POS^ tumors indicated higher tumor uptake for the site-specifically labeled Nb. However, high variation in uptake was observed, which was likely due to variations in the tumor model. Based on these data, an additional experiment was designed involving 42 randomized animals (21/group). Tumor uptake at 80 min. p.i. of the site-specifically and randomly labeled Nbs in the hPD-L1^POS^ 624-MEL tumors was not significantly different, thereby statistically proving the absence of a relevant difference. These results indicate that the position of the NOTA chelator does not negatively impact the affinity, and that the lysine in the CDR does not present a problem for this particular Nb. 

This was, to our knowledge, the first time that the role of a random coupling on lysines was investigated using a Nb with a lysine in its CDR. Our observations are only valid for this ^68^Ga-NOTA labeling, but the effects on affinity and biodistribution might be different if this Nb would be coupled while using lysines to other molecules (e.g., chelator, prosthetic groups, fluorescent dye) or other radioisotopes, as the accessibility of the lysine in the CDR and the effect on binding properties might be very different. Moreover, lysines in the framework structure of Nbs can also participate in the binding, therefore when the crystal structure of the lead Nb is not available, a side-by-side comparison is necessary for such coupling and labeling methods. Nbs with lysines in their CDRs should not be immediately excluded for lead compound selection and for random conjugation, but should rather be evaluated, especially when no information on their crystal structure is available. 

The here reported tumor uptake values were in the same range as for previously pre-clinically tested anti-PD-L1 small-sized PET probes. For example, 2.4 ± 0.3%ID/g uptake was reported for the ^18^F-labeled Adnectin in L297 xenografts at 90 min. p.i. [26] and 1.7 to 5%ID/g for the ^68^Ga-labeled Nb109 at 1 h p.i. in A375 and MCF-7 xenografts, respectively [27]. Tumor-to-blood ratios were in the same range as other probes (6 vs. 5 vs. 10 vs. 3 vs. 5 for site-specifically ^68^Ga-labeled Nb, randomly labeled Nb 80 min. p.i., ^89^Zr-labeled Atezolizumab mAb 72 h p.i. [11], ^18^F-labeled Adnectin 90 min. p.i. [26], and ^68^Ga-labeled Nb109 [27], respectively). Tumor-to-muscle ratios were even higher for the here tested compounds than those that were reported for other anti-PD-L1 PET probes (34 and 28 for our labeled Nbs, compared to only 2–10 for the mAb, 12 for the Adnectin, and 9 for the Nb109) adding to the potential of our compounds. As compared with the ^68^Ga-labeled Nb109, the biological half-life of our Nb variants were shorter (12.4 vs. 10.8 vs. 49.8 min. for the site-specifically ^68^Ga-labeled Nb, randomly labeled Nb and ^68^Ga-labeled Nb109, respectively), as well as lower kidney retention resulting in images with low background. Low-level background activity is especially important to correctly assess low-levels of expression, as it is the case for PD-L1, with for example cut-offs of only 1 to 5% positive cells in immunohistochemistry for optimal treatment selection in lung carcinoma patients [33]. 

Taking together all of the results obtained for each of the functionalization strategies, their high production yields, high purity, specific tumor targeting, and excellent tumor-to-background ratios make them both excellent candidates for future clinical translation. The site-specific coupling method offers the advantage that it yields a homogenous pharmaceutical product, which is not the case for the random lysine conjugation. Its implementation in a GMP radiopharmacy is however less straightforward as compared with the random strategy due to the use of an enzyme. Nevertheless, the use of enzymes in pharmaceutical productions is increasing, for example in antibody drug conjugate production for clinical trials, and such a development might also facilitate the use of the Sortase strategy for radiopharmaceutical production [34]. Since the in vivo study showed no impact on the uptake of the randomly radiolabeled Nb in positive tumors as compared with the site-specifically radiolabeled Nb and, given their similar K_D_ in vitro, the random strategy is currently the easiest strategy for clinical translation in the case of this hPD-L1 Nb. This strategy was also applied to the anti-HER2 and -MMR Nbs that are currently undergoing clinical trials, so that the translation of the hPD-L1 Nb should be similar and straightforward. The radiation burden for the hPD-L1 Nb will be considerably lower when compared with other Nbs in clinical trials due to it low kidney retention. Moreover, as the use of Nbs targeting PD-L1 enables PET imaging already at early time points after injection with patient procedures very similar to current clinical practice with [^18^F]FDG, implementation in routine patient care should also be very straightforward.

## 5. Conclusions

In this study, we have confirmed the high affinity and specificity of a lead Nb targeting human PD-L1 as well as fast clearance in vivo of the ^68^Ga-labeled probe. We demonstrated that a lysine present in the CDR did not impact the affinity when randomly functionalizing this particular Nb with NOTA for ^68^Ga-labeling as compared with a site-specific strategy. Although the site-specific strategy had the advantage of providing a more homogeneous clinical product, the random strategy is more straightforward and less costly for clinical translation. The ^68^Ga-labeled Nb is a promising PET imaging agent for future clinical assessment of PD-L1 expression and patient stratification. When compared to other PET probes in development, the radiolabeled Nb shows the lowest kidney uptake, enables imaging at early time points after injection, and provides high tumor-to-muscle and tumor-to-blood ratios in a preclinical model, thereby confirming its high potential for successful clinical applicability in the future. 

## 6. Patents

Bridoux, J.; Broos, K.; Lecocq, Q.; Raes, G.; Keyaerts, M.; Devoogdt, N.; Breckpot, K.; Xavier, C. Human PD-L1-BINDING Immunoglobulins. hPD-L1 Nanobody No. PCT/EP2019/055133; WO/2019/166622; 6 September 2019. 

Keyaerts, M.; Devoogdt, N.; Raes, G. Radio-labelled Antibody Fragments for Use in the Prognosis, Diagnosis of Cancer as well as for the Prediction of Cancer Therapy Response. HER2-Nb imaging and therapy No. PCT/EP2015/067424; WO/2016/016329; 4 February 2016.

## Figures and Tables

**Figure 1 biomolecules-10-01388-f001:**
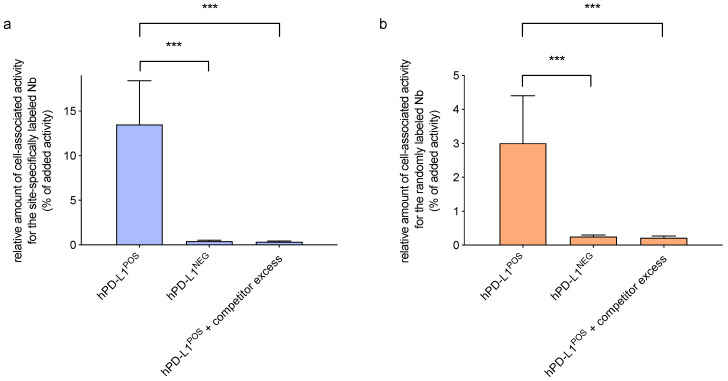
Relative amount of cell-associated activity of the (**a**) site-specifically and (**b**) randomly labeled [^68^Ga]Ga-NOTA-(hPD-L1) Nbs on hPD-L1^POS^ cells at a 3 nM Nb concentration, or on hPD-L1^NEG^ cells, or in presence of an excess of unlabeled Nb as control groups. (***, *p* < 0.001).

**Figure 2 biomolecules-10-01388-f002:**
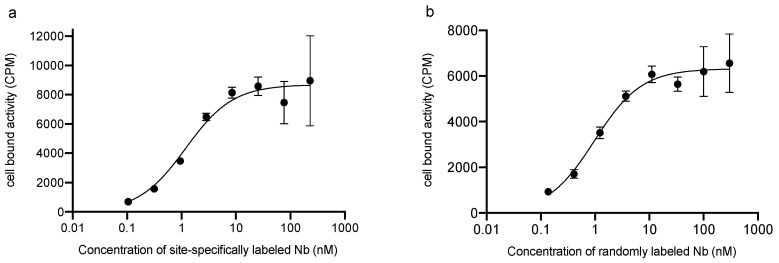
Radioligand binding study on PD-L1^POS^ 624-MEL cells. Cell bound activity in counts per minute (CPM) expressed as a function of the Nb concentration (nM) for (**a**) the site-specifically ^68^Ga-labeled NOTA-(hPD-L1) Nb and (**b**) the randomly ^68^Ga-labeled NOTA-(hPD-L1) Nb.

**Figure 3 biomolecules-10-01388-f003:**
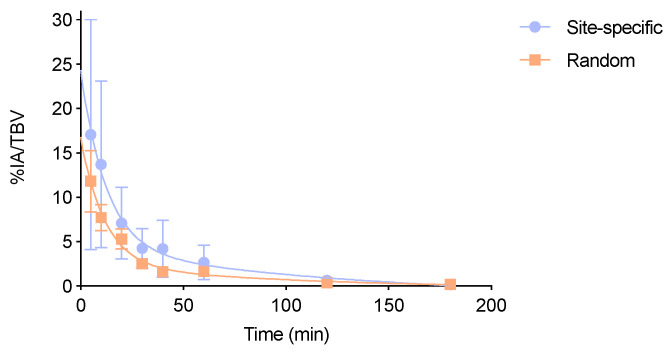
One phase decay fitting curve representing the % of injected activity (IA) per total blood volume (TBV) over time for the site-specifically and randomly labeled [^67^Ga]Ga-NOTA-(hPD-L1) Nbs, showing a biological half-life of 12.4 min. and 10.8 min., respectively (NS).

**Figure 4 biomolecules-10-01388-f004:**
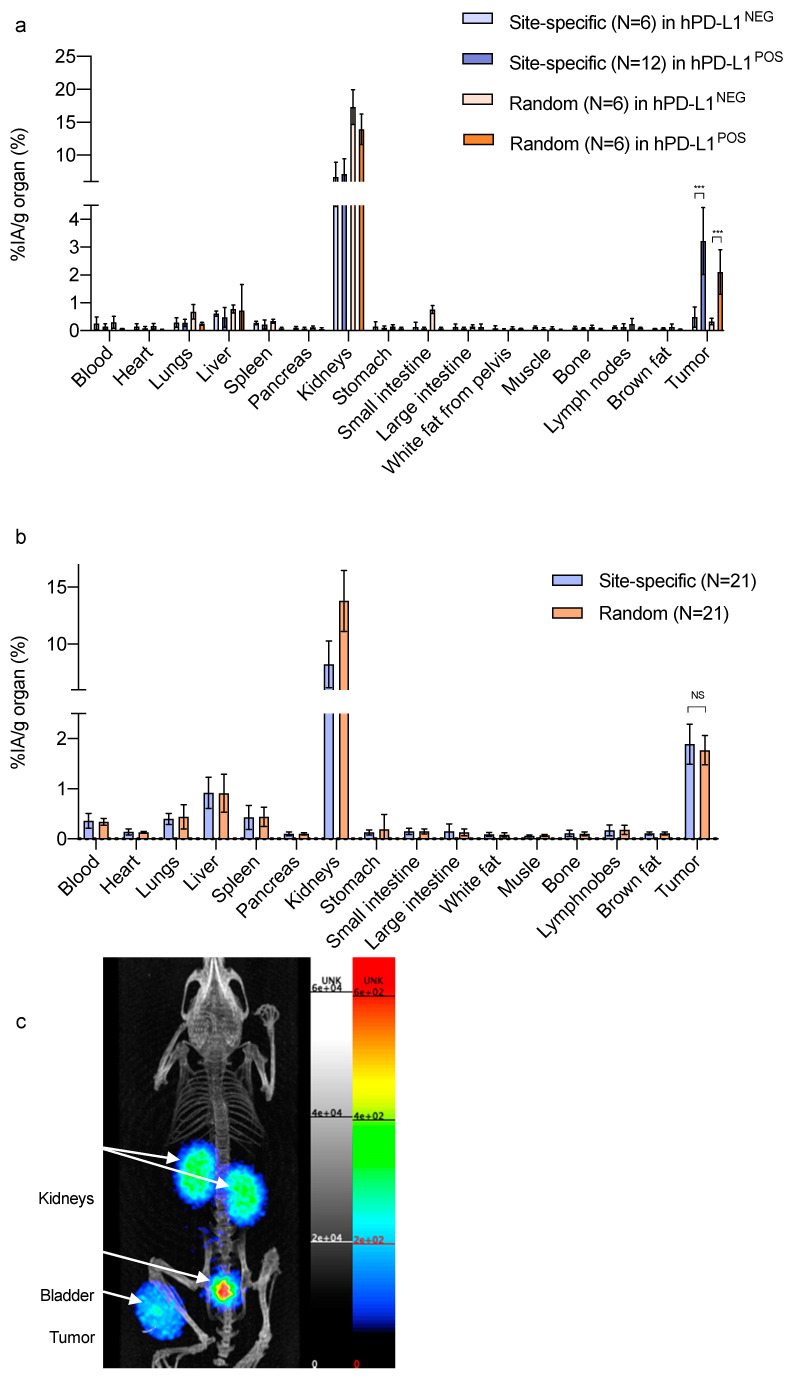
In vivo biodistribution at 1 h 20 p.i. (**a**) Preliminary experiment: Ex vivo biodistribution of site-specifically and randomly labeled [^68^Ga]Ga-NOTA-(hPD-L1) Nbs in hPD-L1^POS^ tumors or hPD-L1^NEG^ tumors. (*** *p* < 0.001) (**b**) Comparison experiment: Ex vivo biodistribution profiles and tumor targeting of site-specifically and randomly labeled [^68^Ga]Ga-NOTA-(hPD-L1) Nbs in hPD-L1^POS^ tumors. (NS, non-significant) (**c**) PET/CT image of a mice bearing hPD-L1^POS^ tumor 1 h 20 p.i. of [^68^Ga]Ga-NOTA-(hPD-L1) Nb, obtained on the β-CUBE PET/CT system. Scale on the PET image is in kBq/mL.

**Table 1 biomolecules-10-01388-t001:** Affinity kinetics measurements by SPR. Equilibrium dissociation constant (K_D_), association constant (k_a_) and dissociation constant (k_d_) of the different Nb variants binding to immobilized recombinant hPD-L1 as determined by SPR: the hPD-L1 Nb containing both the sortag and hexahistidine tags (Nb-sortag-His_6_), the site-specifically coupled NOTA-(hPD-L1) Nb (ss NOTA-Nb), the unmodified hPD-L1 Nb with a histag (Nb-His_6_) and the randomly coupled NOTA-(hPD-L1) Nb (rdm NOTA-Nb).

Nb	k_a_ (1/Ms)	k_d_ (1/s)	K_D_ (nM)
Nb-His_6_	3.98 ± 0.01 × 10^5^	1.49 ± 0.01 × 10^−3^	3.75 ± 0.02
rdm NOTA-Nb	4.35 ± 0.02 × 10^5^	1.59 ± 0.02 × 10^−3^	3.66 ± 0.05
Nb-sortag-His_6_	3.47 ± 0.01 × 10^5^	1.51 ± 0.02 × 10^−3^	4.36 ± 0.06
ss NOTA-Nb	3.34 ± 0.01 × 10^5^	1.47 ± 0.02 × 10^−3^	4.41 ± 0.06

**Table 2 biomolecules-10-01388-t002:** Stability expressed as radiochemical purity (RCP) of the ^67/68^Ga-labeled NOTA-(hPD-L1) Nbs. NOTA-(hPD-L1) Nbs site-specifically labeled with ^68^Ga or ^67^Ga, or randomly labeled with ^68^Ga, in human serum (HS) at 37 °C or in injection buffer (IB) at room temperature (RT) or 40 °C. Lowest RCP value obtained from radio-SEC (*) or radio-iTLC is reported. Time points that were not measured are noted as “NM”.

Time Point (min)	HS (37 °C)			IB (RT)			IB (40 °C)
	Site-Specific		Random	Site-Specific		Random	Site-Specific
	RCP(%) ^68^Ga	RCP(%) ^67^Ga	RCP(%) ^68^Ga	RCP(%) ^68^Ga	RCP(%) ^67^Ga	RCP(%) ^68^Ga	RCP(%) ^68^Ga
0	>99	>99	97	>99	>99	97	>99
30	NM	>99	97	NM	>99	97	NM
60	95 *	98	93 *	NM	>99	96	97 *
120	NM	NM	NM	NM	NM	96	96 *
160	86 *	96 *	NM	NM	NM	NM	NM
220	NM	NM	91 *	>99	NM	95 *	94 *
265	81 *	NM	NM	99	NM	NM	NM
1020	NM	NM	NM	NM	98 *	NM	NM

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
