# Peer review of "Anti-Human PD-L1 Nanobody for Immuno-PET Imaging: Validation of a Conjugation Strategy for Clinical Translation"

_biomolecules, 2020, doi:10.3390/biom10101388_

Round 1

Reviewer 1 Report

see attachment

Author Response

Dear, thank you for the relevant comments. Please find the attached response document.

Reviewer 2 Report

The manuscript "anti-human PD-L1 Nanobody for immuno-PET imaging; validation of a conjugation stratgey for Clinical translation, by Bridoux et al, presents a well performed chemistry-radiochemistry-preclinical study of a highly interesting nanobody.

The chosen strategy to evaluate random versus specific labelling is of importance considering translation to a clinical setting avoiding enzymes in the production. The preclinical results show very promising binding affinities as well as favourable kinetics and biodistribution of both nanobodies. The gallium-68 nanobody is therefore a promising candidate for clinical studies. 

The experimental work is very well performed, described and documented. 

This manuscript can be published as it is.

Author Response

Dear,

Thank you very much for taking the time to review the manuscript and for the comments.

With kind regards

Reviewer 3 Report

The study by Bridoux et al. (Manuscript ID biomolecules-941329) describes the validation of a conjugation strategy for small-sized antibody fragments exemplified with an anti-human PD-L1 nanobody. The authors compare single domain antibody fragments, which have been functionalized with NOTA chelators either by site-specific or random modification, with regard to their binding characteristics and kinetics, their in vitro and in vivo stability as well as regarding their biodistribution and in vivo tumor targeting.

The manuscript is very well written, properly structured and reads very well. I can clearly suggest acceptance based on some minor revision. My suggestions on how this manuscript still can be improved are the following:

The authors state several times (abstract, discussion, conclusion) that random functionalization of lysine residues does not influence the affinity, binding behavior, tumor uptake and biodistribution of the nanobodies. However, such a strong general statement would require much more supporting data using several nanobodies with different specificity and a range of tumor models. I suggest toning down and re-wording that message! Furthermore, it would be of interest, if the authors investigated, to which particular lysine residue the p-NCS-Bn-NOTA was coupled.

Although the authors determined the equilibrium dissociation constant (KD) of the unmodified and NOTA-functionalized nanobodies by surface plasmon resonance measurements and cell binding assays, the immunoreactive fraction (Lindmo et al., J Immunol Methods 1984;72:77-89. [PubMed: 6086763]) or target-binding fraction (Sharma et al., Nucl Med Biol 2019;71:32-38. [PubMed: 31128476]) was not quantified. The authors should add a respective set of data to support the main basic statement of their publication.

The authors used for their study PD-L1 positive as well as a PD-L1 negative cell lines. However, the expression of the target was not confirmed in the manuscript. I suggest providing a corresponding Western Blot analysis for the different cell lines as well as immunohistochemistry / immunofluorescence data of tumor sections to prove the presence of PD-L1 in vitro and in vivo.

In Figure S3, the authors show SDS-PAGE and Western blot analysis of the different modified nanobodies. The approximate molecular weight details of the used protein ladder should be included in the figures. Furthermore, Figure S3c is supposed to be a SDS gel. However, it looks like a Western Blot membrane. Please double-check!

In Figure 2, there are no error bars shown. I assume that the values for these graphs were measured at least in triplicate, so please calculate the standard deviation and add the error bars. For the Y-axis, the authors used cell bound activity (CPM) as units. However, the amount of the bound radioligand (pmol) per milligram of protein (cell lysate) would be much more relevant here. In addition, the calculation of the total number of binding sites Bmax would be helpful here.

The figure legend of Figure 4 contains twice the caption (b). Please correct! Why was the image taken 80 min p.i. , which seems to be a rather uncommon time point. A direct comparison of PET/CT images taken from animals with PD-L1 negative tumors would be of great interest here! Furthermore, PET/CT images of PD-L1 positive tumor bearing animals after blocking with an excess of unlabeled nanobodies would be very illustrative.

Finally, it should also be noted that the manuscript in its current form contains some slang terms (e.g. blood curves?) and lacks some spacing between numbers and units (e.g. 4h and 3mL/min). All numbers up to twelve should be spelled out (six weeks old, two groups, three lysines, etc. instead of 6 weeks old, 2 groups, 3 lysines).

Altogether, this manuscript has only minor problems and deficits in the present form, is in my opinion up to the standard of Biomolecules and should be considered for publication.

Author Response

Dear, thank you for taking the time to review the manuscript and for the relevant comments. Please find attached a response document.
